# Ladder Use in Older People: Type, Frequency, Tasks and Predictors of Risk Behaviours

**DOI:** 10.3390/ijerph18189799

**Published:** 2021-09-17

**Authors:** Cameron Hicks, Erika M. Pliner, Stephen R. Lord, Daina L. Sturnieks

**Affiliations:** 1Falls, Balance and Injury Research Centre, Neuroscience Research Australia, Randwick, NSW 2031, Australia; c.hicks@neura.edu.au (C.H.); s.lord@neura.edu.au (S.R.L.); d.sturnieks@neura.edu.au (D.L.S.); 2Department of Biomedical Engineering, University of Florida, Gainesville, FL 32611, USA; 3School of Medical Sciences, UNSW, Sydney, NSW 1466, Australia

**Keywords:** accidental falls, aged, risk, behaviour, risk factors, safety

## Abstract

Ladder fall and injury risk increases with age. People who present to a hospital after an injurious ladder fall have been surveyed, but little is known about ladder use in the community. The purpose of this study was to: (1) document salient factors related to ladder safety, and (2) determine physical, executive function, psychological and frequency-of-use factors associated with unsafe ladder use in older people. One hundred and two older people (aged 65+ years) were recruited. Participants completed questionnaires on demographics, health, and ladder use (type, frequency, task, behaviours) and underwent assessments of physical and executive function ability. Results showed both older men and women commonly use step ladders (61% monthly, 96% yearly), mostly inside the home for tasks such as changing a lightbulb (70%) and decorating (43%). Older men also commonly use straight ladders (27% monthly, 75% yearly), mostly outside the home for tasks such as clearing gutters (74%) and pruning trees (40%). Unsafe ladder use was more common in males and individuals with greater ladder use frequency, greater quadriceps strength, better upper limb dexterity, better balance, better stepping ability, greater self-reported everyday risk-taking, a lower fear of falling, and fewer health problems compared to their counterparts (all *p* < 0.05). These findings document ladder use by older people and provide insight into unsafe ladder behaviours that may be amenable to interventions to reduce ladder falls and associated injuries.

## 1. Introduction

Domestic ladder falls are a common occurrence in older people with incidence rates reported as high as 2.13 per 1000 inhabitations/year in this age group [1]. While less common than a fall at the same level (e.g., slip on a wet surface), most ladder falls occur from heights greater than one metre and in consequence, even non-fatal falls result in moderate to severe injuries [2,3,4,5]. Further, the risk of fatality and the level of injury severity is greater for older people [6,7]. A report from NSW, Australia, found that 8496 hospital admissions due to domestic ladder falls cost the health system an estimated $51.8 million between 2010 and 2014 [8]. Many ladder falls in older people are due to ladder misuse and could potentially be avoided by the adoption of simple safety measures [3,4]. These safety measures can be implemented through public health support and policy, similar to general fall prevention strategies for older people [9]. Therefore, ladder falls in older people are a serious health issue in need of public health support and policy.

Previous studies of ladder fall risk undertaken in Australia and Sweden have surveyed people who presented to emergency departments or were admitted to a hospital after an injurious ladder fall. These studies report older men are more likely to experience an injurious fall; that most injurious falls involve straight and step ladders and that injurious falls are often due to improper ladder setup or misuse (e.g., overreaching) [3,4,10,11]. Factors associated with injurious ladder falls include complacency, impulsiveness, impatience, being task focused, and convictions of self-sufficiency [3,4]. Climbers with such attitudes and tendencies may select inappropriate ladders, overreach when on a ladder or poorly set up or secure ladders.

The above studies have provided valuable insights into ladder falls but have limitations in that they focused only on falls resulting in serious injuries requiring hospital care. This neglects non-serious injuries, accident precursors and near-misses that have the potential to result in severe injury under a different set of circumstances [12]. Investigating the potential for injury is critical to fully understand severity risk and to further improve safety margins [12]. In addition, previous studies have not documented the frequency of ladder use or investigated the numerous ways older people use and misuse ladders in the domestic setting. This information is required to understand fall risk and develop educational material for safe ladder use.

The purpose of this study was to: (1) document salient factors related to ladder safety in older people (ladder use frequency, tasks and locations; ladder types used; concern about falling from a ladder; self-rated ladder climbing ability and reported ladder use behaviours), and (2) determine demographic, physical, executive function, psychological, and frequency-of-use characteristics associated with unsafe ladder use. Based on published survey findings pertaining to serious ladder falls [2,3,4,10,11] and our previous studies addressing individual factors that influence task performance on step and straight ladders [13,14], we hypothesised that older men would report more unsafe ladder behaviours than older women and that those with better physical function, poorer executive function, reduced fear of falling, and more frequent ladder use would report more unsafe ladder behaviours.

## 2. Materials and Methods

### 2.1. Participants and Public Involvement

One hundred and two community-dwelling people living in the Sydney Metropolitan Area aged 65 years and older were recruited through advertisements, community presentations, volunteer registries, and word-of-mouth between April and September 2018. Participants were excluded if they were not living independently in the community, used a mobility aid inside the home, had a neurological disorder (e.g., Parkinson’s disease, multiple sclerosis, dementia/Alzheimer’s disease), or were unable to complete everyday ladder tasks due to pain. This study was designed without public involvement. The results were disseminated to the participants involved. The Human Research Ethics Committee at the University of New South Wales granted approval for the study (HC17864) and written informed consent was obtained prior to participation.

### 2.2. Demographic, Health, and Ladder Use Questionnaires

Prior to attending the laboratory, participants completed online/postal questionnaires regarding demographics (age, sex), anthropometrics (height, weight), and health (fear of falling, disability, risk-taking behaviour). At the laboratory visit, participants completed an online ladder use questionnaire. Fear of falling was assessed using the Iconographical Falls Efficacy Scale, comprising a 10-item survey with a 4-point response option (not at all concerned, somewhat concerned, fairly concerned, very concerned), scaled 1 to 4, where a higher score denoted a greater fear of falling [15]. Disability was assessed using the WHO Disability Assessment Schedule 2.0, comprising 40 items with a 5-point response option (none, mild, moderate, severe, extreme or cannot do), scaled 1 to 5, where a higher score denoted greater disability [16]. Risk-taking behaviour was assessed using the Everyday Risk-Taking Scale, comprising 10 items with a 4-point response option (never, occasionally, mostly, always), scaled 1 to 4, where a higher score denoted greater everyday risk-taking [17]. We also administered a 38-item questionnaire seeking information on the type of ladders used (step, straight or fixed ladder), frequency of ladder use, ladder use tasks, ladder use locations, concern about falling from a ladder, self-rated ladder climbing ability, and ladder use behaviours.

The ladder use questionnaire comprised 6 questions related to reported ladder use (frequency, tasks, location, and climbing height), concern about falling from a ladder and climbing ability (indicated in Table 1 and Appendix A), and 32 questions related to ladder use behaviours. Frequency of ladder use was measured using a 7-item scale for step, straight, and fixed ladders (daily, few times a week, weekly, monthly, few times a year, once a year and never). Ladder use tasks were measured using 7 check box items (changing a light bulb, cleaning the gutters, washing the windows, cutting branches or picking fruit, getting objects from the attic, decorating, other). Multiple options were able to be selected. Ladder use location was measured using 4 check box items (inside, outside, at home, places other than my home). Multiple options were able to be selected. Concern about falling from a ladder was measured using a 4-item scale (not concerned, slightly concerned, moderately concerned, and very concerned). Self-rated ladder climbing ability was measured using a 5-item scale (below average, slightly-below average, average, slightly above average, above average).

The 32 ladder use behaviour questions were a series of yes or no questions inquiring about safe, questionable, and unsafe ladder use behaviours. Of these, 11 questions reflected unsafe ladder use behaviours (indicated in Table 2). The sum of unsafe ladder use behaviours was used to create an unsafe ladder behaviour score for each participant (range from 0 to 11), where a higher score signified higher risk for unsafe ladder use. Questions with accompanying photos demonstrating ladder behaviours were provided for the ladder use questions. Certain questions were not asked for fixed ladders (e.g., those relating to changing a light bulb, pruning trees, facing away from the ladder) as such situations and behaviours were considered unlikely.

### 2.3. Vision, Physical, and Executive Function Assessments

During a single laboratory visit to Neuroscience Research Australia (NeuRA) in Sydney, Australia, participants were assessed for vision, physical performance, and executive function by a trained Research Assistant. Visual contrast sensitivity was assessed with the Melbourne Edge Test, measured as the lowest contrast patch correctly identified on a non-grating contrast sensitivity chart with 20 circular patches [18]. Physical performance was assessed from quadriceps strength, coordinated stability, Loop and Wire Test, and Choice Stepping Reaction Time Test. Quadriceps strength was a measure of lower limb strength measured as the maximal (from three trials) isometric knee extension force (kg) with participants seated, knee flexed to 90 degrees and a custom-built strain gauge attached to the lower leg, where a higher force denoted greater strength [18]. Coordinated stability measured controlled pelvis movement from the error score received when guiding a stylus through a maze using movement about the waist, where a higher score denoted worse coordinated stability [19]. The Loop and Wire test measured upper limb unilateral movement and dexterity as the number of wire touches that occurred when guiding a loop through a twisting wire, where a greater number of touches denoted worse upper limb unilateral movement and dexterity [20]. The Choice Stepping Reaction Time test measured stepping ability from the time to react to one of four target stimuli and complete a step, where greater time denoted worse stepping ability [21]. Executive function was assessed using the Trail-Making Test parts A and B, subtracting the time taken to complete test A from time taken to complete test B to characterise processing speed, attention, and task shifting, where a greater time denoted worse executive function performance [22].

### 2.4. Statistical Analysis

The sample size was chosen to have sufficient participants for group comparisons outlined in this paper and conduct multivariable models addressing task performance on step and straight ladders in companion papers [13,14]. Reported ladder use frequency, tasks, and locations are presented in tabular form. Participants who were unable to safely complete a physical assessment due to physical incapacity were given a score of three standard deviations above or below the group mean to reflect their poor performance on that assessment. Participants who scored worse than this had their score censored at this level. Predictor variables for 13 of 102 participants (totalling 17 data points) were also found to have errors or were missing. For the latter cases, these data were retrieved through five iterations of multiple imputation that were based on all other individual factors. Based on a near-median split of unsafe ladder behaviour scores, participants were categorised into safe (0–3 unsafe behaviours) and unsafe (4 or more unsafe behaviours) ladder use groups. Chi-squared tests for contingency tables were used to examine ladder use frequency (at least monthly and yearly), sex, concern about falling from a ladder, and perceived ladder use ability between safe and unsafe ladder user groups. Independent sample t-tests were conducted to examine whether there were significant differences in demographic, anthropometric, executive function, psychological, health/disability, and physical test measures between the safe and unsafe ladder user groups. To permit parametric analyses with the remaining continuously-scored predictor variables, data with right-skewed distributions were logarithmic or square root transformed. The analyses were performed with statistical software (IBM SPSS, Version 26. IBM Corp., Armonk, NY, USA) and significance levels were set at *p* < 0.05.

## 3. Results

Assessed participants comprised 102 people (51 women) with a mean age of 72.9 (±5.5) years, who were relatively healthy as indicated by a mean WHO Disability Assessment Schedule 2.0 score of 5.42 (SD = 5.57). 

Step ladders were the most used ladders with 61% and 96% of participants using these ladders at least monthly and yearly, respectively. Step ladders were most often used inside the home (86%) and most commonly for changing a lightbulb (70%), house decorating (43%), and washing windows (35%). Fifteen percent of participants reported using a straight ladder at least monthly and 57% reported using a straight ladder at least yearly. Straight ladders were mostly used outside the home (90%) for tasks such as clearing gutters (62%), washing windows (33%), and pruning tree branches/picking fruit (38%). Fixed ladders were used infrequently for activities such as accessing an attic (26%), swimming pool 37%) or boat (26%). Two participants (2%) reported no ladder use in the past year. Men and women were equally likely to report using step ladders (χ^2^ = 0.16, df = 1, *p* = 0.685) at least once per month, while men were more likely to use straight ladders at least once per month (χ^2^ = 13.31, df = 1, *p* < 0.001). There was also a trend indicating more men than women used fixed ladders at least once per year (χ^2^ = 3.52, df = 1, *p* = 0.061). Detailed information on ladder use (frequency, tasks, locations, and climbing heights) for men, women, and the total sample are presented in Table 1.

Most participants reported their ladder use ability as “average” or “better than others” across step, straight, and fixed ladders (Table 2). When participants were asked “How concerned are you of falling from a ladder?”, 38% of participants indicated they were not concerned, 41% indicated they were slightly concerned, 10% indicated they were moderately concerned, and 11% indicated they were very concerned.

Participants’ reported ladder use behaviours for the two most used ladder types (step and straight ladders) are presented in Table 2. Of the unsafe ladder behaviours, “climbing with items in hands” was the most common, reported by 75% of step ladder users and 71% of straight ladder users. Other commonly reported unsafe ladder behaviours were ladder use without someone holding the ladder (61%), standing higher than the recommended step height for straight ladders (61%) and step ladders (43%), and overreaching while on a ladder (34%). Less commonly reported behaviours were standing with only one foot on a ladder (22%), standing on toes while on a ladder (15%), climbing a step ladder without hand support (13%), climbing more than one step at a time on a straight ladder (11%) or step ladder (9%), descending while not facing the step ladder (4%) or straight ladder (4%), moving a ladder by hopping on the ladder (4%), and standing on a ladder with someone else on the ladder (3%).

Based on their unsafe ladder behaviour scores, 57 people were classified as safe ladder users and 44 as unsafe ladder users. More men (*n* = 28, 54.9%) than women (*n* = 16, 34%) were classified as unsafe ladder users (χ^2^ = 5.39, df = 1, *p* = 0.027). The unsafe ladder user group were more likely to have little or no concern about falling from a ladder (χ^2^ = 6.963, df = 1, *p* = 0.008) and above average ladder climbing ability for step (χ^2^ = 4.755, df = 1, *p* = 0.029), straight (χ^2^ = 414.419, df = 1, *p* < 0.001), and fixed ladders (χ^2^ = 8.101, df = 1, *p* = 0.004). The unsafe ladder group were also more likely to report using step (χ^2^ = 4.96, df = 1, *p* = 0.040) and straight ladders (χ^2^ = 6.35, df = 1, *p* < 0.022) at least once per month, and fixed (χ^2^ = 10.69, df = 1, *p* = 0.0001) ladders at least once per year, compared to the safe ladder user group.

Table 3 presents age, anthropometric and disability measures, and performance scores for the physical performance and executive function measures for the two ladder user groups. The unsafe group had fewer health problems, less disability, better quadriceps strength, better upper limb dexterity, better coordinated stability, better stepping ability, reduced fear of falling, and reported more risk-taking behaviours than the safe group (*p* < 0.05). Trail-Making and visual contrast sensitivity test scores, age, height, and weight did not differ significantly between the ladder user groups (*p* > 0.05).

## 4. Discussion

This study provides descriptive information on ladder use, including ladder type, frequency and tasks, and examines predictors of reported unsafe ladder use. These findings indicate that both older men and women commonly use step ladders, often inside the home for tasks such as changing a lightbulb and retrieving objects from a height. Older men also commonly use straight ladders outside the home for tasks such as clearing gutters, washing windows, and pruning trees. Unsafe ladder behaviour was also common, with 44% of the cohort reporting four or more unsafe ladder behaviours. The most common unsafe ladder behaviours were climbing while holding items, climbing a ladder without someone else holding onto it, standing higher than the recommended step height, and overreaching when standing on a ladder.

Participants categorised as unsafe ladder users were more likely to be men, have fewer health and disability problems, use ladders more frequently, have better quadriceps strength, better upper limb dexterity, better stepping ability, greater coordinated stability, less fear of falling, and a greater likelihood of everyday risk-taking than those categorised as safe ladder users. These findings identify three overarching factors associated with reported unsafe ladder use: better physical ability, more experience with ladders, and less concern of falling/greater likelihood to take risks in daily life.

In companion papers to this work, we found participants with better physical ability, as determined by performances in tests of upper limb function, quadriceps strength, coordinated stability, and processing speed were faster in undertaking household tasks on step [13] and straight [14] ladders than their less physically- able counterparts. Therefore, it seems that those more physically capable in performing ladder tasks are prepared to accept more risk trade-offs [23]. This is also likely the case for more regular ladder users who may consider they are well practiced and competent in ladder climbing.

In relation to less concern of falling/greater likelihood to take risks in everyday life, our findings complement research undertaken with people who require care in hospital settings due to an injurious ladder fall, of which more than 80% are men [4]. Among these patients, complacency (e.g., no concern about ladders being a problem due to repeated use), impulsiveness/impatience (e.g., not having time to wait for a co-worker and feeling fit and able enough to do it themselves), distraction and being task-focused (e.g., thinking ahead to the job at hand rather than climbing the ladder) as well as having sub-optimal knowledge of safe ladder use (e.g., not securing the ladder, climbing with an item in their hands or climbing with bare feet) were found to be key factors leading to ladder falls [3,4].

While distraction is linked to ladder falls, we did not find poor executive function, as measured by the Trail-Making Test, to be associated with reported unsafe ladder behaviour. This is likely due to the sample being a relatively healthy one without executive dysfunction. Similarly, the lack of association between vision and ladder behaviour likely reflects the lack of participants with poor vision in our sample.

We acknowledge some study limitations. The study was conducted at a single site with the survey administered to a relatively small convenience sample. While the participants comprised both men and women, with a wide age range, living within a range of dwellings from apartments to free-standing homes with large gardens, it is acknowledged some of our findings may not generalize beyond similar locales, and that larger studies are required to provide definitive findings regarding robust rates and proportions of unsafe ladder use. Participants may have had other health conditions or impairments that were not explicitly queried (for example, hearing impairment) and while all participants were living independently in the community, it was not recorded whether they were living in a house, unit or apartment. Second, it could be considered that the item pertaining to climbing with items in the hand is an inherent part of many ladder-climbing tasks. However, the US Occupational Health and Safety Administration recommend against climbing with items in the hand [24], and as strategies to avoid this are possible (securing the item in a waist-belt, having a second person pass, and receiving items from the climber once in position, etc.), we feel the inclusion of this item in our unsafe ladder behaviour scale is warranted. Finally, this study focused on ladder use behaviours while the person was on the ladder and did not investigate ladder use setup. Improper ladder setup is another commonly reported factor that is associated with ladder falls [3,4] and warrants attention in future studies.

Public health support and policy have the potential to reduce ladder falls. Education promoting safe ladder use has been suggested as a way of reducing ladder fall injury [4,8,25,26], and several general campaigns have been run by government bodies [27,28] yet with untested effects. The Australian Competition & Consumer Commission released a Ladder Safety Matters National Campaign in 2016. The campaign combined a website with information on risks and injuries, buying tips and safe use, a range of patient story videos and flyers and posters. It targeted men over the age of 55 and focused on both the consequences of ladder falls and simple solutions to preventing a fall [27]. More targeted campaigns, such as publication in relevant magazines and hardware stores may be more effective [4,25,26]. Further, ladder users would likely benefit from policy at the local, state, and federal levels that protect them from unnecessary risks. In addition, related research has shown the wearing of safety helmets significantly reduces intracranial injuries from work-related falls up to 4 m in height [29], so the promotion of this simple initiative may also assist in the prevention of fall-related head injuries in the home setting. Our research suggests education and policy should address specific behaviours including avoiding climbing with items in hand, not using ladders alone, and following ladder use recommendations regarding step height and reaching. Safe ladder education could also be included in healthy ageing and fall prevention initiatives for older men, such as the inclusive, male-friendly Men’s Sheds program [30]. In this setting, qualitative research has shown such initiatives can influence older men to adopt a more cautious mindset, and pay greater attention to potential fall risks, be careful, concentrate, and slow down [30].

## 5. Conclusions

This study describes ladder use type, frequency, tasks, and behaviour in a sample of healthy older people living in the community. Results suggest that older people who report more unsafe ladder behaviours tend to be male, use ladders more frequently, have fewer health and disability problems, better quadriceps strength, better upper limb dexterity, better stepping ability, better coordinated stability, a lower fear of falling, and a greater propensity for everyday risk-taking than those who report fewer unsafe ladder behaviours. Our findings suggest that interventions to reduce ladder fall injuries should target risk-takers, who are more likely to be male, with good physical function, and low fear. Such interventions could include educational programs and policy related to the risks of ladder falls and recommendations for safe ladder use.

## Figures and Tables

**Table 1 ijerph-18-09799-t001:** Participants’ reported ladder use ability **^#^**.

	Step Ladder	Straight Ladder	Fixed Ladder
Below Average	5 (4.9)	18 (17.6)	23 (22.5)
Slightly Below Average	12 (11.8)	16 (15.7)	13 (12.7)
Average	48 (47.1)	42 (41.2)	43 (42.2)
Slightly Above Average	19 (18.6)	13 (12.7)	9 (8.8)
Above Average	18 (17.6)	13 (12.7)	14 (13.7)

**^#^** Participants rated their ladder climbing ability “compared to others”.

**Table 2 ijerph-18-09799-t002:** Number (%) of participants who reported “yes” in relation to their use of step, straight, and fixed ladders.

Question	Step Ladder [*n* = 96]	Straight Ladder [*n* =56]
Common for me to		
Descend facing the ladder	94 (97.9)	54 (96.4)
Descend not facing the ladder ^+^	4 (4.2)	2 (3.6)
Climb without using my hands ^+^	12 (12.5)	0 (0)
Climb with only one hand on the ladder	32 (33.3)	21 (37.5)
Climb with both hands on the ladder	67 (69.8)	51 (91.1)
Climb one step at a time	72 (75.0)	38 (67.9)
Climb more than one step at a time ^+^	9 (9.4)	6 (10.7)
Climb without items in my hands	43 (44.8)	31 (55.4)
Climb with items in my hands ^+^	72 (75.0)	40 (71.4)
Climb with a tool belt	11 (11.5)	8 (14.3)
I climb		
A-frame ladders	60 (62.5)	N.A.
Step stool ladders (3 or less steps)	84 (87.5)	N.A.
On A-frame or straight ladder I have		
Stood higher than the recommended step height *^+^	41 (42.7)	34 (60.7)
Stood on the top cap/rung of the ladder	15 (15.6)	9 (16.1)
Straddled the ladder	13 (13.5)	N.A.
On step stool ladders		
The guardrail was not present	36 (37.5)	N.A.
I held on to the guardrail	37 (38.5)	N.A.
I did not hold on to the guardrail	15 (15.6)	N.A.
	**Across all Ladders [*n* = 99]**	
Common for me to stand		
With feet planted	92 (92.9)	
On my toes ^+^	15 (15.2)	
With only one foot on the ladder ^+^	22 (22.2)	
With my body in the middle of the ladder	82 (82.8)	
On the ladder while overreaching ^+^	34 (34.3)	
Without someone holding the ladder ^+^	60 (60.6)	
With someone holding the ladder	47 (47.5)	
With someone else on the ladder ^+^	3 (3.0)	
I have moved a ladder by		
Hopping on the ladder ^+^	4 (4.0)	
Climbing down and picking it up	91 (91.9)	
I have climbed a ladder when I was		
Fatigued	14 (14.1)	
Tired	27 (27.3)	
Dizzy	0 (0)	
Sick	2 (2.0)	

* stepladders = no higher than the second highest step (not including the top cap as a step); straight ladders = no higher than 4th highest step; ^+^ items classified as unsafe ladder behaviours; *n* = the number of participants who responded to these items.

**Table 3 ijerph-18-09799-t003:** Age and anthropometric, executive function, psychological, health/disability, and physical test measures for the safe and unsafe ladder user groups.

Variable	Safe Ladder Users Mean (SD)	Unsafe Ladder Users Mean (SD)	
Age	73.5 (6.1)	72.9 (4.6)	t_99_ = 1.433 *p* = 0.155
Height	1.69 (1.04)	1.71 (0.84)	t_99_ = −1.304 *p* = 0.195
Weight	71.7 (14.4)	73.6 (11.9)	t_99_ = −0.707 *p* = 0.481
Trail-Making Test (B-A) (s)	41.7 (20.6)	44.2 (29.1)	t_99_ = −0.008 *p* = 0.994
Iconographical Fall Efficacy Scale	15.4 (3.8)	12.2 (2.7)	t_99_ = 5.220 *p* < 0.001
Everyday Risk-Taking Scale	22.8 (4.8)	26.7 (4.0)	t_99_ = −4.348 *p* < 0.001
WHO Disability Assessment Schedule 2.0	7.00 (6.45)	3.36 (3.33)	t_99_ = 3.150 *p* = 0.002
Visual contrast sensitivity (dB)	23.3 (1.3)	23.4 (1.2)	t_99_ = 0.807 *p* = 0.422
Loop and Wire Test (errors)	24.8 (13.6)	17.8 (10.7)	t_99_ = 2.503 *p* = 0.014
Quadriceps Strength Test (kg)	32.8 (13.8)	38.8 (11.4)	t_99_ = −2.575 *p* = 0.011
Coordinated Stability Test (error score)	5.8 (7.0)	2.6 (3.7)	T98.9 = 2.687 *p* = 0.008
Choice Stepping Reaction Time Test (s)	1.08 (0.14)	1.03 (0.10)	T98.7 = 2.291 *p* = 0.024

High scores in the Trail-Making, Loop and Wire, Coordinated Stability, and Choice Stepping Reaction Time tests, and low scores in the Visual contrast sensitivity and quadriceps strength tests indicate impaired performances. Higher scores in the Concern about ladder falls, Iconographical Fall Efficacy Scale, Everyday Risk-Taking and WHO Disability Assessment scales indicate greater concern, fear, risk, and disability respectively.

## Data Availability

Data are available on reasonable request.

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
