# Peer review of "Ladder Use in Older People: Type, Frequency, Tasks and Predictors of Risk Behaviours"

_ijerph, 2021, doi:10.3390/ijerph18189799_

Round 1

Reviewer 1 Report

This is a well-designed and small-scale survey. I only have two comments. 

First, please justify the determination of sample size. Typically, the sample size is somewhat inadequate to support specific analyses and cannot obtain reliable rates or proportions.

Second, the presentations of all tables could be improved to make them easy to read, such as clarifying the purpose of each table in the title, explaining critical numbers in the tables using notes below the table. Table 1 includes too many numbers and is hard to read. 

Author Response

  1. First, please justify the determination of sample size. Typically, the sample size is somewhat inadequate to support specific analyses and cannot obtain reliable rates or proportions.

The sample size was chosen to have sufficient participants for group comparisons outlined in this paper and conduct multivariable models addressing task performance on step and straight ladders in companion papers (Pliner et al., 2021; Pliner et al., 2020). We acknowledge that the sample may not provide robust rates or proportions for all ladder use behaviours, it does provide important insight into domestic unsafe ladder use and provides valid comparisons between safe and unsafe ladder user groups with respect to physical ability, executive function and behavioural and demographic characteristics. We have described the sample size choice in the methods and expanded our limitation section of the discussion regarding the limited sample size as below.

Statistical Analysis, paragraph 1
The sample size was chosen to have sufficient participants for group comparisons outlined in this paper and conduct multivariable models addressing task performance on step and straight ladders in companion papers (Pliner et al., 2021; Pliner et al., 2020).

Discussion, paragraph 6
The study was conducted at a single site with the survey administered to a relatively small convenience sample. While the participants comprised both men and women, with a wide age-range, living within a range of dwellings from apartments to free-standing homes with large gardens, it is acknowledged some of our findings may not generalize beyond similar locales, and that larger studies are required to provide definitive findings regarding robust rates and proportions of unsafe ladder use.

  1. Second, the presentations of all tables could be improved to make them easy to read, such as clarifying the purpose of each table in the title, explaining critical numbers in the tables using notes below the table. Table 1 includes too many numbers and is hard to read.

The titles of the tables have been simplified. Table 1 has been moved to supplementary material with an improved layout to make it easier to read.

Reviewer 2 Report

The authors had conducted a topic of interest related to ladder use in older people. I have carefully read the manuscript and found it difficult to read and understand the whole picture of the study. There were several issues related to logic, measures, analysis, interpretations, and presentations. I found that the analysis was not sophisticated enough to answer all the research questions and hypotheses. Hope the following comments could be helpful for future submission.

Abstract

Please add the time and place the study was conducted.

Please add the test results to elaborate on the associations.

The results seem to be wrongly interpreted “Those who reported more unsafe ladder behaviors were more likely to be male, use ladders frequently, have better balance and stepping ability, and report more everyday risk-taking, a lower fear of falling and fewer health problems, compared to those who reported fewer unsafe ladder behaviors.” I guess the outcome should be “unsafe ladder use”, NOT the gender, ladder frequency use, balance and stepping ability, everyday risk-taking, fear of falling, or health problems. Instead, these should be the predictors of “unsafe ladder use”. Please readdress these. It should be “…, as compared to those were female, less frequently ladder use, worse physical function, less everyday risk-taking, higher fear, and more health problems” as their counterparts.

Introduction

Please indicate the location (which country) for ladder fall incidence.

There are inconsistent terms used in the last part of the introduction and those in the findings in the Abstract. Please correct them and use them consistently.

Materials and Methods

How did you decide to have a sample size of 102? Which sampling technique was used?

Please provide the IRB granted number.

Please describe more clearly about each questionnaire/measure, e.g., how many items per scale, response options (yes, no; or Likert scale), how to score or categorize them. It’s better to write one variable/group of variables in one paragraph.

How were the questionnaires administered? E.g., self-administered by participants or interviewed by the researcher or research assistants?

Was “executive function” the cognitive function? If so, please address this.

Which machine/equipment was used to assess visual contrast sensitivity, quadriceps strength?

Who did conduct the visual, physical, and cognitive assessment?

Please provide the evidence to support the cut-off value of unsafe ladder use.

If you treat “unsafe ladder use” as a dependent variable. Then why did you examine the effects of unsafe ladder use risk groups and sex on demographic, physical, cognitive, and psychological measures?

Chi-squared test was used for categorical variables, which test was used to conduct the parametric analyses?

Results

Please shorten up the title of Table 1.

Authors analyzed and reported the results by different strata of ladder types, gender, but you did not explain in the purpose or statistical part. Please readdress this.

The categories of ladder use ability were not described previously. How did you classify them into 5 groups? Please address this in the methods.

Do you have statistics for “fixed ladder” in Table 3? If not, please explain this in the methods.

It is not clear to read the results of Table 4 if you present them in the current format. It causes confusion and loss of the link to the purpose of the study.

Which tests were used in Table 4? Please mark and note to clearly explain. Remember to make sure the “unsafe ladder use” is an outcome variable, not an independent variable in the test.

Author Response

  1. Please add the time and place the study was conducted.

We have now added this information to the methods section.

Materials and methods, paragraph 1
One hundred and two community dwelling people living in the Sydney Metropolitan Area aged 65 years and older were recruited through advertisements, community presentations, volunteer registries and word-of-mouth between April and September 2018.

Vision, physical and executive function assessments, paragraph 1
During a single laboratory visit to Neuroscience Research Australia (NeuRA) in Sydney, Australia

  1. Please add the test results to elaborate on the associations.

We have indicated in the abstract that all associations are significant at p <0.05. We have also added the t-test values and degrees of freedom in table 4.

  1. The results seem to be wrongly interpreted “Those who reported more unsafe ladder behaviors were more likely to be male, use ladders frequently, have better balance and stepping ability, and report more everyday risk-taking, a lower fear of falling and fewer health problems, compared to those who reported fewer unsafe ladder behaviors.” I guess the outcome should be “unsafe ladder use”, NOT the gender, ladder frequency use, balance and stepping ability, everyday risk-taking, fear of falling, or health problems. Instead, these should be the predictors of “unsafe ladder use”. Please readdress these. It should be “…, as compared to those were female, less frequently ladder use, worse physical function, less everyday risk-taking, higher fear, and more health problems” as their counterparts.

Thank you. We have modified the abstract text to read:

Male sex, frequent ladder use, good quadriceps strength, upper limb dexterity, balance and stepping ability, low fear of falling, high everyday risk-taking and few health problems were identified as significant predictors of unsafe ladder use.

  1. Introduction
    Please indicate the location (which country) for ladder fall incidence.

We have now added this detail into the introduction section.

Introduction, paragraph 2
Previous studies of ladder fall risk undertaken in Australia and Sweden

  1. There are inconsistent terms used in the last part of the introduction and those in the findings in the Abstract. Please correct them and use them consistently.

We believe we have updated all our terms to be consistent throughout the paper. For example, we have changed older adults to older people, risky ladder behaviour to unsafe ladder behaviour and cognition to executive function Please note fear of falling (based-off the Iconographical Falls Efficacy Scale) is a different measure than concern about falling from a ladder (a question asked in the ladder use survey).

Materials and Methods
How did you decide to have a sample size of 102? Which sampling technique was used?

Please see response to Reviewer 1, point 1.

  1. Please provide the IRB granted number.

We have now added this detail into the methods section.

Materials and Methods, paragraph 1
The Human Research Ethics Committee at the University of New South Wales granted approval for the study (HC17864)

  1. Please describe more clearly about each questionnaire/measure, e.g., how many items per scale, response options (yes, no; or Likert scale), how to score or categorize them. It’s better to write one variable/group of variables in one paragraph.

We have now included this information in the methods section.

Demographic, health and ladder use questionnaires, paragraph 1, 2 and 3

Fear of falling was assessed using the Iconographical Falls Efficacy Scale, comprising a 10-item survey with a 4-point response option (not at all concerned, somewhat concerned, fairly concerned, very concerned), scaled 1 to 4, where a higher score denoted a greater fear of falling (Delbaere et al., 2011). Disability was assessed using the WHO Disability Assessment Schedule 2.0, comprising 40-items with a 5-point response option (none, mild, moderate, severe, extreme or cannot do), scaled 1 to 5, where a higher score denoted greater disability (Üstün et al., 2010). Risk-taking behaviour was assessed using the Everyday Risk-Taking Scale, comprising 10-items with a 4-point response option (always, mostly, occasionally, never), scaled 1 to 4, where a lower score denoted greater everyday risk-taking (Butler et al., 2015). We also administered a 38-item questionnaire seeking in-formation on the type of ladders used (step, straight or fixed ladder), frequency of ladder use, ladder use tasks, ladder use locations, concern about falling from a ladder, self-rated ladder climbing ability and ladder use behaviours.

The ladder use questionnaire comprised 6 questions related to reported ladder use (frequency, tasks, location, and climbing height), concern about falling from a ladder and climbing ability (indicated in Table 1 and Supplementary 1) and 32 questions related to ladder use behaviours. Frequency of ladder use was measured using a 7-item scale for step, straight and fixed ladders (daily, few times a week, weekly, monthly, few times a year, once a year and never). Ladder use tasks were measured using 7 check box items (changing a light bulb, cleaning the gutters, washing the windows, cutting branches or picking fruit, getting objects from the attic, decorating, other). Multiple options were able to be selected. Ladder use location was measured using 4 check box items (inside, outside, at home, places other than my home). Multiple options were able to be selected. Concern about falling from a ladder was measured using a 4-item scale (not concerned, slightly concerned, moderately concerned and very concerned). Self-rated ladder climbing ability was measured using a 5-item scale (below average, slightly-below average, average, slightly above average, above average).

The 32 ladder use behaviour questions were a series of yes or no questions inquiring about safe, questionable and unsafe ladder use behaviours. Of these, 11 questions reflected unsafe ladder use behaviours (indicated in Table 2). The sum of unsafe ladder use behaviours was used to create an unsafe ladder behaviour score for each participant (range from 0 to 11), where a higher score signified higher risk for unsafe ladder use. Questions with accompanying photos demonstrating each behaviour were provided for step and straight ladder specific questions and general ladder use questions.

  1. How were the questionnaires administered? E.g., self-administered by participants or interviewed by the researcher or research assistants?

We have now added this information in the methods section.

Demographic, health and ladder use questionnaires, paragraph 1
Prior to attending the laboratory, participants completed an online/postal questionnaire regarding demographics (age, sex), anthropometrics (height, weight) and health (fear of falling, disability, risk-taking behaviour). At the laboratory visit participants completed an online ladder use questionnaire.

  1. Was “executive function” the cognitive function? If so, please address this.

We now only refer to executive function throughout the manuscript. This includes the description of the Trail Making Tests.

Vision, physical and executive function assessments, paragraph 1
Executive function was assessed using the Trail-Making Test parts A and B, subtracting the time taken to complete test A from time taken to complete test B to characterise processing speed, attention and task shifting [21].

  1. Which machine/equipment was used to assess visual contrast sensitivity, quadriceps strength?

We have now added these details into the methods section.

Vision, physical and executive function assessments, paragraph 1
Visual contrast sensitivity was assessed with the Melbourne Edge Test, measured as the lowest contrast patch correctly identified on a non-grating contrast sensitivity chart with 20 circular patches (Lord et al., 2003). Physical performance was assessed from quadriceps strength, coordinated stability, Loop & Wire Test, and Choice Stepping Reaction Time Test. Quadriceps strength was a measure of lower limb strength measured as the maximal (from three trials) isometric knee extension force (kg) with participants seated, knee flexed to 90 degrees and a custom built strain gauge attached to the lower leg, where a higher force denoted greater strength (Lord et al., 2003). Coordinated stability measured con-trolled pelvis movement from the error scored received when guiding a stylus through a maze using movement about the waist, where a higher scored denoted worse coordinated stability (Lord et al., 1996). The Loop & Wire test measured upper limb unilateral movement and dexterity as the number of wire touches that occurred when guiding a loop through a twisting wire, where a greater number of touches denoted worse upper limb unilateral movement and dexterity  (Ingram et al., 2019). The Choice Stepping Reaction Time test measured stepping ability from the time to react to one of four target stimuli and complete a step, where greater time denoted worse stepping ability (Schoene et al., 2017). Executive function was assessed using the Trail-Making Test parts A and B, subtracting the time taken to complete test A from time taken to complete test B to characterise pro-cessing speed, attention and task shifting, where a greater time denoted worse executive function performance (Tombaugh, 2004).

  1. Who did conduct the visual, physical, and cognitive assessment?

We have now added these details into the methods section.

Vision, physical and executive function assessments, paragraph 1
During a single laboratory visit to Neuroscience Research Australia (NeuRA) in Sydney, Australia participants were assessed for vision, physical performance and executive function by a trained Research Assistant.

  1. Please provide the evidence to support the cut-off value of unsafe ladder use.

This is the first publication describing this cut-off value. We based it on a near median cut point of the sum of unsafe ladder use behaviours. This is now described more clearly in the methods section.

Statistical analyses, paragraph 1
The low and high risk for unsafe ladder use groups were determined by creating dichotomous groups based on a near median cut point of the sum of unsafe ladder use behaviours.

  1. If you treat “unsafe ladder use” as a dependent variable. Then why did you examine the effects of unsafe ladder use risk groups and sex on demographic, physical, cognitive, and psychological measures?

We have now clarified throughout the manuscript that our objective was to characterise unsafe ladder users and identify associations of unsafe ladder use and not the effects of unsafe ladder use risk groups and sex on demographic, physical, cognitive, and psychological measures.

  1. Chi-squared test was used for categorical variables, which test was used to conduct the parametric analyses?

We used group t-tests – this in now indicated in the statistical analysis section of the methods.

Statistical Analysis, paragraph 1
Chi-squared tests for contingency tables were used to examine ladder use frequency (at least monthly and yearly), sex, concern about falling from a ladder, and perceived ladder use ability between safe and unsafe ladder user groups. Grouped t-tests were conducted to examine whether there were significant differences in demographic, anthropometric, executive function, psychological, health/disability and physical test measures between the safe and unsafe ladder user groups.

Results

  1. Please shorten up the title of Table 1.

The title of this table (now Supplementary table 1) has been simplified.

Supplementary Table 1. Participant reported step, straight and fixed ladder use (frequency, tasks, locations and climbing heights) for men, women and the total sample (Number of responses and percentages).

  1. Authors analyzed and reported the results by different strata of ladder types, gender, but you did not explain in the purpose or statistical part. Please readdress this.

We have now simplified the findings and removed the gender comparisons.

  1. The categories of ladder use ability were not described previously. How did you classify them into 5 groups? Please address this in the methods.

We have now added this detail into the methods section.

Demographic, health and ladder use questionnaires, paragraph 2
Self-rated ladder climbing ability was measured using a 5-item scale (below average, slightly-below average, average, slightly above average, above average).”

  1. Do you have statistics for “fixed ladder” in Table 3? If not, please explain this in the methods.

The questions asked for step and straight ladder use are specific to these types of ladders. Given the different designs of these ladders, individuals may use them differently (e.g. more people may climb down step ladders like stairs – facing away from the ladder than on straight ladders). Some of the questions asked for step ladders were not asked for straight ladders as these questions do not apply to straight ladder use (e.g. holding onto the guardrail for step stool ladders). Similarly, many of the questions asked for straight ladder use are unlikely to be as relevant for fixed ladder use. Specifically, many fixed ladders are installed vertically, this makes the questions, such as, descending the ladder facing away from the ladder and climbing without using ones hands very difficult and unlikely. Questions that we felt were common to multiple ladders (including fixed ladders), regardless of design, are the questions asked for all ladders. However, some of the all ladder questions are only relevant to step and straight ladders (e.g. moving the ladder and someone else moving the ladder).

We have added to the methods to indicate this.

Demographic, health and ladder use questionnaires, paragraph 3
Questions with accompanying photos demonstrating ladder behaviours were provided for the ladder use questions. Certain questions were not asked for fixed ladders (e.g. those relating to changing a light bulb, pruning trees, facing away from the ladder) as such situations and behaviours were considered unlikely.

  1. It is not clear to read the results of Table 4 if you present them in the current format. It causes confusion and loss of the link to the purpose of the study.

We have now simplified the findings presented in the table (new Table 3) and removed the gender comparisons.

  1. Which tests were used in Table 4? Please mark and note to clearly explain. Remember to make sure the “unsafe ladder use” is an outcome variable, not an independent variable in the test.

We apologize for the confusion. New Table 3 now clearly indicates that unsafe ladder use risk group is the outcome variable. It is also now indicated in the statistical analysis section of the methods that grouped t-tests were conducted to examine whether there were significant differences in demographic, anthropometric, executive function, psychological, health/disability and physical test measures between the safe and unsafe ladder user groups.

Statistical Analysis, paragraph 1
Grouped t-tests were conducted to examine whether there were significant differences in demographic, anthropometric, executive function, psychological, health/disability and physical test measures between the safe and unsafe ladder user groups.

Reviewer 3 Report

Comments to the manuscript: ijerph-1331379 titled: “Ladder use in older people: type, frequency, tasks and predictors of risk behaviours”. Authors have carried out a research using a questionnaire to older people using differents common ladders.

Minor changes are required and review the language in the manuscript.

ABSTRACT

Is redacted in proper manner

KEYWORDS

Are Mesh term?

INTRODUCTION

Can authors reference the economical cost in the society by ladder falls injuries?

MATERIAL AND METHODS

Line 69: Can authors explain hoy calculate the sample size?

Line 72-73: Did authors keep in mind subjects with hearing disturbances?

RESULTS

Line 128: Please, correct the word “Heathy”

Table 4: Add  in the table the term “Variable”. Please explain the numbers between paragraphs in the table. Please add “p value”

DISCUSSION

Line 219: Can authors explain what mean “+4”?

Line 223-228: Can authors explain better this paragraph?

Line 250-263: Please include subjects with hearing disturbances. All subjects lived in the same type of housing?

Line 264-273: Could author recommend a campaign based in previous studies adapted to their research based in your findings

CONCLUSION

Is adapted to the objetive of the study and redacted in proper manner.

Author Response

  1. Keywords

Are Mesh term?

We have used MESH terms as well as “ladder use”, as this is a prime focus of our paper.

  1. INTRODUCTION
    Can authors reference the economic cost in the society by ladder falls injuries?

We have now included in the introduction data from a report from NSW, Australia found that 8496 hospital admissions due to domestic ladder falls cost the health system an estimated $51.8 million between 2010 and 2014 (Miu et al., 2016).

  1. MATERIAL AND METHODS
    Line 69: Can authors explain how calculate the sample size?

Please see response to reviewer 1, point 1.

  1. Line 72-73: Did authors keep in mind subjects with hearing disturbances?

Poor hearing was not an exclusion criterion, but all participants had adequate hearing to understand test instructions and complete all assessments. Please also see our response to the latter question 32 regarding the proportion of people with hearing impairment.

  1. RESULTS
    Line 128: Please, correct the word “Heathy”

This has been corrected. Refer to Results, paragraph 1

  1. Table 4: Add in the table the term “Variable”. Please explain the numbers between paragraphs in the table. Please add “p value”

Table 4 has been updated and p values and the term variable have been added.

DISCUSSION

  1. Line 219: Can authors explain what mean “+4”?

In this sentence 4+ refers to those people who reported four or more unsafe ladder behaviours. For clarity we have elaborated to describe this with four or more.

Discussion, paragraph 1
Unsafe ladder behaviour was also common, with 44% of the cohort reporting four or more unsafe ladder behaviours.

  1. Line 223-228: Can authors explain better this paragraph?

This paragraph has been redrafted as follows.

Results, paragraph 6
Table 3 presents age, anthropometric and disability measures and performance scores for the physical performance and executive function measures for the two ladder user groups. The unsafe group had fewer health problems, less disability, better quadriceps strength, better upper limb dexterity, better coordinated stability, better stepping ability, reduced fear of falling, and reported less concern about falling from a ladder and more risk-taking behaviours than the unsafe group (p<0.05). Trail Making and visual contrast sensitivity test scores and age did not differ significantly between the ladder user groups (p>0.05).

  1. Line 250-263: Please include subjects with hearing disturbances. All subjects lived in the same type of housing?

We have now added this detail into the discussion.

Discussion, paragraph 6
Participants may have had other health conditions or impairments that were not explicitly queried (for example hearing impairment) and while all participants were living independently in the community it was not recorded whether they were living in a house, unit or apartment.

  1. Line 264-273: Could author recommend a campaign based in previous studies adapted to their research based in your findings

We have now added this detail into the discussion.

Discussion, paragraph 7
The Australian Competition & Consumer Commission released a Ladder safety matters national campaign in 2016. The campaign combined a website with information on risks and injuries, buying tips and safe use, a range of patient story videos and flyers and posters. It targeted men over the age of 55 and focussed on both the consequences of ladder falls and simple solutions to preventing a fall (Australian Competition and Consumer Commission. Ladder Safety Matters National Campaign., 2016).

Round 2

Reviewer 2 Report

Dear Authors,

Thanks for addressing all the comments.

One point related to the name of the test (Group t-test) that you should revise. There is no such test name in biostatistics. Please consult with a statistician for the correct name.

Looking forward to reading the article online soon.

Best,

Author Response

Thank you, we have changed the term group t-test to independent samples t-test.

Statistical Analysis, paragraph 1
Independent samples t-tests were conducted to examine whether there were significant differences in demographic, anthropometric, executive function, psychological, health/disability and physical test measures between the safe and unsafe ladder user groups.